# FOXD1 and Gal-3 Form a Positive Regulatory Loop to Regulate Lung Cancer Aggressiveness

**DOI:** 10.3390/cancers11121897

**Published:** 2019-11-28

**Authors:** Chien-Hsiu Li, Yu-Chan Chang, Michael Hsiao, Shu-Mei Liang

**Affiliations:** 1Graduate Institute of Life Sciences, National Defense Medical Center, Taipei 114, Taiwan; dicknivek@icloud.com; 2Agricultural Biotechnology Research Center, Academia Sinica, Taipei 115, Taiwan; 3Genomics Research Center, Academia Sinica, Taipei 115, Taiwan; jameskobe0@gmail.com; 4Department of Biochemistry, College of Medicine, Kaohsiung Medical University, Kaohsiung 80708, Taiwan

**Keywords:** lung cancer, FOXD1, Galectin-3

## Abstract

Dysregulation of forkhead box D1 (FOXD1) is known to promote tumor progression; however, its molecular mechanism of action is unclear. Based on microarray analysis, we identified galectin-3/LGALS3 (Gal-3) as a potential downstream target of FOXD1, as FOXD1 transactivated *Gal-3* by interacting with the *Gal-3* promoter to upregulate Gal-3 in FOXD1-overexpressing CL1-0 lung cancer cells. Ectopic expression of FOXD1 increased the expression of Gal-3 and the growth and motility of lung cancer cells, whereas depletion of Gal-3 attenuated FOXD1-mediated tumorigenesis. ERK1/2 interacted with FOXD1 in the cytosol and translocated FOXD1 into the nucleus to activate *Gal-3*. Gal-3 in turn upregulated FOXD1 via the transcription factor proto-oncogene 1 (ETS-1) to transactivate *FOXD1*. The increase in ETS-1/FOXD1 expression by Gal-3 was through Gal-3-mediated integrin-β1 (ITGβ1) signaling. The overexpression of both FOXD1 and Gal-3 form a positive regulatory loop to promote lung cancer aggressiveness. Moreover, both FOXD1 and Gal-3 were positively correlated in human lung cancer tissues. Our findings demonstrated that FOXD1 and Gal-3 form a positive feedback loop in lung cancer, and interference of this loop may serve as an effective therapeutic target for the treatment of lung cancers, particularly those related to dysregulation of Gal-3.

## 1. Introduction

Lung cancer is one of the most prevalent cancers worldwide [1]. Approximately 80% of lung cancers are non-small cell lung cancers (NSCLCs) [2]. Surgery, chemotherapy, radiotherapy, molecular-targeted therapy, and immunotherapy are the current standard treatments. However, patients with advanced lung cancer have 10–15% overall 5-year survival rates, an median overall survival (OS) of about 11.1 months, and a post-recurrence survival rate (PRS) of 13% (hazard ratio (HR) = 0.78) [2,3,4]. Even with second-line drug treatment, such as nivolumab, progression-free survival (PFS) is about 2.3–3.5 months [5]. It is therefore critical to elucidate the molecular mechanisms of lung cancer proliferation and metastasis to identify and develop novel therapeutic targets in lung cancer.

Transcription factor (TF) dysregulation has been linked to tumor progression, and targeting TFs may be an effective strategy for cancer intervention [6]. FOXD1, also known as *FKHL8* and *FREAC-4*, is a transcription factor that belongs to the forkhead box (FOX) family [7,8,9,10]. FOXD1 is associated with cell programming, especially in renal and kidney development [7,8,9,10]. Abnormal FOXD1 expression is involved in the progression of tumors [11] including breast cancer [12], colorectal cancer, melanoma, glioma [13], osteosarcoma, renal cell carcinoma, ovarian carcinoma, medulloblastoma [14], and lung cancer [15]. Recently, clinical mRNA microarray data identified FOXD1 as a factor associated with poor prognosis and that is required for lung cancer cell proliferation [15]. FOXD1 overexpression is strongly associated with NSCLC proliferation and metastasis through the activation of vimentin [16]. Although several studies have tried to unravel the molecular networks of FOXD1 [11,12,13,16,17], the molecular mechanisms related to FOXD1 remain largely unexamined, especially in lung cancer.

Gal-3 belongs to the β-galactoside-binding lectins family [18]; it is ubiquitously expressed in different cellular compartments, including the cytoplasm, nucleus, cell surface, and extracellular matrix [19,20]. Gal-3 has pleiotropic functions ranging from gene splicing, cell–cell interactions, and immune responses, to embryonic development [19,20,21,22]. Elevated Gal-3 contributes to tumor progression in different human cancers [23,24,25,26]. Extracellular Gal-3 acts as a mediator that regulates metastatic cancer processes by recognizing N-acetyllactosamine (β-galactoside)-conjugated proteins, including EGFR, integrins, TGF-β receptors, etc. [25,27,28]. For instance, the interaction of integrin β3 and Gal-3 contributes to EGFR tyrosine kinase inhibitor resistance, which facilitates lung cancer cell stemness properties and renders them more resistant to chemotherapeutic agents [29]. The interaction between Gal-3 and β-catenin is required for drug resistance and stemness in lung cancer [21]. Therefore, several studies have searched for potential strategies by targeting Gal-3 [19,25,26,30]. These studies showed that Gal-3 expression is regulated by bFGF, NGF-mediated MEK/ERK signaling, and various transcription factors, such as Runx family proteins, HIF-1α, and NF-κB [30]. Whether there is any interrelationship between FOXD1 and Gal-3 remains unclear.

In this study, we demonstrated that FOXD1 bound to the *Gal-3* promoter and positively transactivated *Gal-3* expression. Moreover, ERK1/2 was required for FOXD1 translocation into the nucleus by associating with FOXD1. Notably, the ERK signaling cascade was mediated by the Gal-3/integrin-β1 (ITGβ1) axis. Gal-3 induced FOXD1 expression by associating with the ITGβ1/FAK/PI3K/AKT/PHB1/Ras/Raf-1 complex in lipid rafts and increasing proto-oncogene 1 (ETS-1) to transactivate *FOXD1*. Gal-3 mediated ERK activation, leading to the translocation of FOXD1 into the nucleus to regulate Gal-3 expression. The co-expression of FOXD1 and Gal-3 promoted, whereas the co-knockdown of both attenuated, lung cancer cell growth and motility *in vitro*. Furthermore, using clinical human lung cancer tissue microarrays, we found that both FOXD1 and Gal-3 were positively correlated in advanced lung tumor tissues.

## 2. Results

### 2.1. High FOXD1 and Gal-3 Expression are Associated with the Poor Prognosis in Lung Cancer

To identify novel downstream molecules involved in FOXD1-mediated lung cancer aggressiveness, *FOXD1* cDNA was overexpressed in cells or knocked down by specific siRNA, and mRNA microarrays and comparative gene expression analyses were then performed. We selected the probes in which there was >5-fold change in *FOXD1* knockdown versus vector control in CL1-0 cells. Using Ingenuity Pathway Analyses (IPA), we identified several potential regulators (Figure 1A). Among these regulators, *Gal-3* was shown to be one of the *FOXD1* downstream effectors. In addition, the core-analysis from IPA also revealed that RNA level of *Gal-3* (*LGALS3*) was increased by >100-fold by *FOXD1* upregulation (Figure 1B). Furthermore, both *FOXD1* and *Gal-3* mRNA and protein levels were increased in lung tumor cells, such as CL1-0, CL1-5, and H1299 cells, compared with normal lung cells (WI-38) by qPCR (Figure 1C) and Western blot (Figure 1D) assays. We also found that a high level of *FOXD1* mRNA and protein expression was positively correlated with *Gal-3* expression in more aggressive cells, such as CL1-5 and H1299 (Appendix A). We thus evaluated *FOXD1* and *Gal-3* gene expression in lung cancer patients from the SurvExpress database (LUAD-TCGA and Lung Meta-base) and found high *FOXD1* and *Gal-3* expression in the high-risk group (Figure 1E). Moreover, lung cancer patients with high *FOXD1* and *Gal-3* gene expression were associated with a poor prognosis in a different database (LUAD-TCGA and Lung Meta-base) (Figure 1F). Collectively, these data indicated that FOXD1 might promote lung cancer aggressiveness through the upregulation of *Gal-3*.

### 2.2. FOXD1 is a Transcription Factor of Gal-3

To understand whether FOXD1 upregulated *Gal-3* by directly binding to the *Gal-3* promoter, we searched for the transcription factor binding site in the *Gal-3* promoter sequence from the JASPAR database (http://jaspar.genereg.net/). We found that *Gal-3* promoter contains the FOXD1 binding sequences (TCCATAGTTTACATAAG). A luciferase promoter assay was then performed to confirm the prediction. As shown in Figure 2A, *Gal-3* promoter activity was enhanced 5.8-fold by ectopic FOXD1 expression compared to without FOXD1 stimulation (*p* < 0.01). Moreover, mutation of the FOXD1 binding motif from TAGTTTAC to TAACCTGC decreased FOXD1-mediated *Gal-3* promoter activity. To examine whether FOXD1 can directly bind to the *Gal-3* promoter, an ChIP-qPCR assay was performed. FOXD1 was found to bind to the promoter region (−1075 to −1058 nt) of the *Gal-3* gene in human lung cancer cells (Figure 2B). To further verify whether FOXD1 could act as an upstream factor to regulate *Gal-3* expression, FOXD1 was overexpressed by *FOXD1* cDNA or knocked down by specific siRNA. Both qPCR and Western blot assays revealed that FOXD1 overexpression resulted in the upregulation of *Gal-3* (Figure 2C); in contrast, FOXD1 depletion resulted in the downregulation of *Gal-3* (Figure 2D). Furthermore, we observed that the overexpression of FOXD1 increased the proliferation and colony-forming ability of lung cancer cells, while the depletion of *Gal-3* attenuated the phenotypes induced by FOXD1 (Figure 2E,F). Moreover, increased cancer cell migration and invasion ability are consistent with the FOXD1-overexpression models (Figure 2G). These results indicated that FOXD1 transactivates *Gal-3* expression to promote lung cancer aggressiveness. 

Recently, FOXD1 has been identified in both the cytosol and the nucleus [13]. To further determine which molecules control FOXD1 translocation into the nucleus to regulate Gal-3, we used Group-based prediction system (GPS). As FOXD1 has two ERK interaction sites (T135 and S141) in its nuclear localization signal (Appendix A), we suspected that its translocation might be controlled by ERK1 and/or ERK2. Immunofluorescence staining revealed that both ERK and FOXD1 localized to the cytosol and the nucleus of CL1-5 cells (Appendix A). Since ERK has been reported to activate the expression of different genes by associating with different transcription coactivators, such as Runx1 to regulate Gal-3 expression [25,30], we examined whether ERK is involved in FOXD1-mediated Gal-3 expression. Using an ERK inhibitor (PD98059) to reduce ERK phosphorylation, we showed that inhibition of the phosphorylation of ERK decreased the expression levels of FOXD1 and Gal-3 (Appendix A). The knockdown of *ERK1* or *ERK2* by siRNA in CL1-5 cells suppressed FOXD1 and Gal-3 expression (Appendix A). Moreover, the inactivation of ERK attenuated FOXD1 translocation into the nucleus (Appendix A). Similar to the ERK inhibitor results, the depletion of ERK1 or ERK2 reduced FOXD1 in the nucleus and retained the level of FOXD1 in the cytosol (Appendix A). To elucidate whether ERK directly interacts with FOXD1, we examined the association between ERK and FOXD1 by treating cells with the ERK inhibitor (PD98059). The immunoprecipitation assay showed that ERK interacted with FOXD1 in both the cytosol and nucleus. The downregulation of phosphorylated ERK^T185/Y187^ (p-ERK) expression (but not ERK) by PD98059 suppressed the interaction of ERK and FOXD1 in the cytosol and nucleus (Appendix A). Similar to the ERK inhibitor results, the interaction of p-ERK and FOXD1 was downregulated by the knockdown of ERK1 or ERK2 (Appendix A). These results suggest that ERK1/2 interacts with FOXD1 in both the cytosol and nucleus, and p-ERK is required for FOXD1 translocation into nucleus.

To further determine whether ERK is directly involved in FOXD1 binding to the *Gal-3* promoter, we performed a luciferase assay to measure the *Gal-3* promoter luciferase activity upon the suppression of p-ERK. As shown in Appendix A, FOXD1-mediated *Gal-3* luciferase activity was markedly decreased by treating cells with the ERK inhibitor PD98059 (left) or by the depletion of ERK1 or ERK2 (right) (*p* < 0.001). Moreover, a ChIP-qPCR assay also revealed that the inactivation of ERK or the knockdown of ERK1 and ERK2 attenuated FOXD1 binding to the *Gal-3* promoter (Appendix A). Functionally, FOXD1-mediated cell proliferation was suppressed upon the inactivation of ERK signaling by inhibitors or by transfection with *ERK1*-siRNA or *ERK2*-siRNA (*p* < 0.01) (Appendix A). In addition, the colony-formation ability promoted by FOXD1 was also suppressed by PD98059 (*p* < 0.001) or the depletion of ERK1/ERK2 (*p* < 0.001) (Appendix A). The FOXD1-mediated migration/invasion was suppressed when CL1-0 cells were treated with an ERK inhibitor or the depletion of ERK1 or ERK2 (*p* < 0.001) (Appendix A). Taken together, we concluded that FOXD1 associated with p-ERK1/2 to translocate into the nucleus and bind to the *Gal-3* promoter through the interaction with FOXD1.

### 2.3. ETS-1 Is Involved in Gal-3-Mediated FOXD1 Expression

To further identify molecules involved in Gal-3-mediated lung cancer aggressiveness, Gal-3 was overexpressed by *Gal-3* cDNA or knocked down by specific siRNA. mRNA microarrays and comparative gene expression analyses were performed. Ingenuity Pathway Analyses showed that *ETS-1* was one of the Gal-3 downstream effectors (Figure 3A–B). Both qPCR and Western blot assays revealed that Gal-3 depletion resulted in the downregulation of *ETS-1* (Figure 3C, left). In contrast, Gal-3 overexpression resulted in the upregulation of *ETS-1* (Figure 3D, right). We next investigated whether ETS-1 regulated FOXD1. We found that the depletion of ETS-1 downregulated FOXD1 at both the mRNA and protein levels in CL1-5 cells (Figure 3E-F, left), whereas the overexpression ETS-1 upregulated FOXD1 at the mRNA and protein levels in CL1-0 cells (Figure 3E–F, right). Moreover, we found that ETS-1-mediated FOXD1 expression contributed to further increase Gal-3 expression (Figure 3E–F). Reporter and ChIP-qPCR assays revealed that FOXD1 mediated the activation of *Gal-3* expression in an ETS-1-dependent manner (Figure 3G–H). These findings suggest that Gal-3 increases *ETS-1* expression that not only upregulates *FOXD1* but also collaborates with FOXD1 to increase FOXD1-mediated activation of Gal-3 expression.

### 2.4. Gal-3 Regulates FOXD1 Expression through ITGβ1 Signaling

As the interaction of Gal-3 with integrins promotes cancer progression, and integrins are one of the ERK upstream effectors [25,31], we next investigated whether FOXD1 expression is regulated by the Gal-3/ITGβ1/ERK axis. First, we investigated whether Gal-3 is regulated FOXD1 through ERK. Gal-3 overexpression upregulated FOXD1 at both the mRNA and protein levels in CL1-0 cells (Figure 4A, left), whereas the depletion Gal-3 by siRNA downregulated FOXD1 at the mRNA and protein levels in CL1-5 cells (Figure 4A right). We then examined whether Gal-3 associates with ITGβ1. We found that Gal-3 associated with ITGβ1/FAK/PI3K/AKT/PHB1/Ras/Raf1 in the lipid rafts. Gal-3 overexpression increased, whereas knockdown of Gal-3 suppressed, the interaction (Figure 4B, left) and the phosphorylation of ITGβ1/FAK/PI3K/AKT/PHB1/Ras/Raf1 in the lipid rafts (Figure 4B, right). To determine whether the interaction of Gal-3 and ITGβ1 is required for FOXD1 expression, we transfected CL1-0 cells with a *Gal-3* expression vector or/and *ITGβ1* siRNA. The overexpression of Gal-3 by an expression vector activated ITGβ1 signaling and FOXD1 expression, whereas the co-transfection of *Gal-3* cDNA and *ITGβ1* siRNA to block ITGβ1 signaling resulted in the downregulation of FOXD1 (Figure 4C, left). Conversely, ectopic ITGβ1 expression increased downstream signaling and upregulated FOXD1 expression. Compared with overexpression of ITGβ1 alone, the co-transfection of the *ITGβ1* expression vector and *Gal-3* siRNA reduced the levels of FOXD1 (Figure 4C, right). Moreover, treating cells with rhGal-3 to mimic exo-Gal-3 inducing ITGβ1 signaling also increased FOXD1 expression (Appendix A). Since ERK1/2 regulates FOXD1 translocation into the nucleus to bind to *Gal-3* promoter (Appendix A), we next performed an immunoprecipitation assay and showed that the activation of ERK increased FOXD1 translocation into the nucleus. As shown in Figure 4D, the overexpression of Gal-3 or ITGβ1 increased the interaction of ERK and FOXD1 in the cytosol and nucleus. We also compared *Gal-3* promoter activity when cells were transfected with Gal-3 or ITGβ1. The reporter and ChIP-qPCR assays revealed that overexpression of Gal-3 or ITGβ1 upregulated *Gal-3* promoter activity (Appendix A). The biological assays showed that FOXD1 depletion attenuated cell growth, as indicated by proliferation/colony formation (Appendix A).

Transwell assays showed that Gal-3-mediated cell migration/invasion ability could be attenuated by FOXD1 depletion in CL1-0 cells (Appendix A). Taken together, these findings suggest that ITGβ1/ERK signaling mediates the relationship between Gal-3 and FOXD1 in lung cancer.

### 2.5. FOXD1 and Gal-3 form a Positive Regulatory Loop to Promote Lung Cancer Cell Growth and Motility

To examine whether FOXD1 and Gal-3 may form a positive regulatory loop to promote tumor cell aggressiveness, gene manipulation of each gene was performed. Our results showed that the ectopic expression of FOXD1 or Gal-3 enhanced proliferation/colony formation and cell migration/invasion in CL1-0 cells (Appendix A). Conversely, the depletion of FOXD1 or Gal-3 decreased proliferation/colony formation and cell migration/invasion in CL1-5 cells (Appendix A). To evaluate the feed-forward loop effects of both FOXD1 and Gal-3 on tumor cells, the cells were transfected with *FOXD1* and *Gal-3* siRNA. The depletion of both FOXD1 and Gal-3 resulted in the downregulation of FOXD1 and Gal-3 in an additive manner (Figure 5A). Functionally, the co-knockdown of *FOXD1* and *Gal-3* further attenuated cell growth and migration/invasion in CL1-5 cells (Figure 5A–C). In contrast, ectopic enforced expression of both FOXD1 and Gal-3 further increased their protein expression (Figure 5D) coupled with enhanced cell growth and migration/invasion in CL1-0 cells (Figure 5D–F). These data demonstrated that FOXD1 and Gal-3 may form a positive regulatory loop to promote lung cancer aggressiveness. 

### 2.6. FOXD1 and Gal-3 are Positively Correlated in Human Lung Cancer Tissues

Accumulating evidence suggests that both FOXD1 and Gal-3 are highly expressed in several cancers [15,16,21], whereas the clinicopathological relationship is still not well understood. To evaluate the correlation between FOXD1 and Gal-3 in lung cancer tissues, we determined FOXD1 or Gal-3 expression by immunochemistry staining in human lung cancer tissue microarrays. FOXD1 and Gal-3 were increased in lung tumor tissues compared with those in lung normal tissues (*p* < 0.001) (Figure 6A,B), increased in T3 + T4-size tumors (*n* = 23) compared with those in T1+T2 size tumors (*n* = 87) (*p* < 0.001), increased in tumors with those in lymph node involvement (N1 + N2, *n* = 53) compared with those in tumors without lymph node involvement (N0, *n* = 57) (*p* < 0.001), and increased in stages II + III + IV (*n* = 56) compared with in stage I (*n* = 54) tumors (*p* < 0.001). Importantly, we also found that a high level of FOXD1 expression positively correlated with Gal-3 expression (*r* = 0.624, *p* < 0.0001, Appendix A). Overall, these data indicated that FOXD1 and Gal-3 form a positive relationship to promote tumor progression in human lung cancer tissues. 

## 3. Discussion

Tumor heterogeneity is one of the reasons for malignancy and treatment failure, including in lung cancer [32]. Insights into the regulation of cancer may provide directions for the development of new therapeutic strategies. In this study, we demonstrated that FOXD1 promoted lung cancer aggressiveness by targeting Gal-3, which acts as an oncogene in lung cancer [26] (Figure 2). Our results show that ERK1/2 regulated Gal-3 by associating with FOXD1 in both the cytosol and the nucleus. (Appendix A). In addition, Gal-3 regulated FOXD1 expression through the ITGβ1/ERK/ETS-1 cascade (Figure 4). FOXD1 and Gal-3 form a positive loop that promotes lung cancer aggressiveness (Figure 5). 

The dysregulation of FOXD1 expression has previously been associated with malignant behavior in certain cancers and has been reported to contribute to aspects of tumor progression, such as drug resistance, metastasis, and stemness [11]. Aberrant *FOXD1* mRNA expression was recognized as an independent marker in NSCLC patients [15]. Consistently, we and others have shown that elevated FOXD1 protein expression can be detected in clinical lung cancer tissues and associated with tumor malignancy (Figure 6) [16]. Ectopic or depleted expression further suggested that FOXD1 promotes the growth and motility of lung cancer cells (Appendix A). FOXD1 expression was also related to the expression of different EMT markers (Appendix A). These results are consistent with those of a previous study in which FOXD1 promoted cell proliferation and motility through vimentin activation in NSCLC [16]. These data, together with the previous report, reveal that targeting FOXD1 may improve interventions in lung cancer. It is well documented that FOXD1 contributes to aggressiveness by targeting different downstream molecules in various tumor types [11]. The tumorigenic role of FOXD1 has been identified; however, the oncogenic role remains controversial in various cancers. For example, in medulloblastoma, FOXD1 acts as a tumor suppressor by targeting NKX2.2. In contrast, in breast cancer, FOXD1 promotes drug resistance by downregulating P27 [12]. In glioma, FOXD1 upregulates ALDH1A3 to promote stemness properties [13]. FOXD1 also targets RAC1B to regulate melanoma cell motility [17]. However, the downstream target involved in FOXD1 in lung cancer remains to be characterized. Here, we provide evidence that links FOXD1 with Gal-3 in lung cancer, in which FOXD1 enhanced Gal-3 expression (Figure 2); subsequently, Gal-3-mediated ITGβ1/ETS-1 signaling contributed to further increase FOXD1 expression (Figure 4). In particular, we identified this feedback loop that likely sustains the aggressiveness of lung cancers.

Gal-3 belongs to chimera type of β-galactoside-binding lectin [18]. Accumulating evidence demonstrates that in lung cancer, elevated Gal-3 is linked to the promotion of tumor progression [21,29]. Consistent with this model, gain- and loss-of function experiments were used to evaluate the promotion of growth and motility of lung cancer cells by Gal-3 (Appendix A). We and others have suggested that Gal-3 expression correlated with lymphatic metastasis and tumor progression in lung cancer patients (Figure 6) [21]. The transcription factor Runx1 has been identified as being involved in Gal-3 expression [30]. However, a previous study reported that the knockdown of RUNX1 increased lung cancer cell growth and motility, and lung cancer patients with low RUNX1 exhibit a poor prognosis [28]. Intriguingly, there is no report of FOXD1 being associated with Gal-3 expression. Notably, we identified that *Gal-3* has a consensus binding sequence of FOXD1 (TAGTTTAC) on the *Gal-3* promoter (−1075 to −1058 nt), which has ability to recruit the FOXD1 interaction (Figure 2). Our data demonstrate that *Gal-3* is transcriptionally regulated by FOXD1 (Figure 2B), thereby promoting metastasis ability (Figure 2G).

More recent studies suggested that FOXD1 localizes to the cytosol and nucleus [13]. However, the mechanism of the regulatory action of FOXD1 translocation has not been determined. Group-based prediction system (GPS) prediction shows that FOXD1 has an ERK association site in nuclear localization signals (NLS). ERK is known to activate the expression of different genes by association with different transcriptional co-activators [30,33,34]. In addition, several reports have shown that ERK signaling acts through association with Runx1 to regulate Gal-3 [25,30]. To our knowledge, this report is the first to show ERK1/2 interaction with FOXD1 in both the cytosol and nucleus (Appendix A), in which the inactivation of ERK or the depletion ERK1 or ERK2 suppresses the translocation of FOXD1 into the nucleus to transactivate *Gal-3* expression (Appendix A). In response to signal transduction, the catalytic domain of the ERK1/2 kinase activity requires dual phosphorylation, and using an inhibitor is not sufficient to discriminate which form of ERK is critical for FOXD1 translocation [35]. Our data suggest that both ERK1 and ERK2 were essential for FOXD1 translocation (Appendix A); therefore, FOXD1-mediated cell growth and motility were regulated by ERK signaling (Appendix A). These results indicate that ERK activation increases the interaction of ERK with FOXD1 in the cytosol, leading to translocation into the nucleus to bind the *Gal-3* promoter and to promote tumor growth and motility.

In this study, we found that FOXD1 expression or translocation was regulated by ERK (Appendix A). Notably, a previous study reported that depletion of Gal-3 led to ERK dephosphorylation in PDAC cells [36]. In this study, for the first time, we found that FOXD1 could also be regulated by Gal-3 (Figure 3 and Figure 4); however, the underlying mechanism involved in Gal-3-mediated FOXD1 was not fully elucidated. In this study, we found that FOXD1 expression was regulated by ETS-1 (Figure 3). As Gal-3 can phosphorylate ERK signaling through the association with different molecules on the cell surface, including ITGβ1 [25,29], it is likely that ERK-mediated FOXD1 translocation can, at least partially, be regulated by the Gal-3-mediated ITGβ1/ERK/ETS-1 cascade. The overexpressing and/or knockdown of Gal-3 in CL1-5 cells showed that Gal-3 regulates the association of ITGβ1/FAK/PI3K/AKT/PHB1/Ras/Raf-1 in lipid rafts (Figure 4); this result is also consistent with our previous report [37]. Our findings validated Gal-3-enhanced FOXD1 expression as being dependent on the presence of ITGβ1 (Figure 4C). In addition, the upregulation of Gal-3 activates ERK and promotes the interaction of FOXD1 and ERK in the cytosol and nucleus (Figure 4D), which in turn activates *Gal-3* expression (Appendix A). Therefore, Gal-3 stimulated proliferation, colony formation, migration, and invasion, and these effects were attenuated by downregulating FOXD1 (Appendix A). These results indicate that ITGβ1/ERK signaling bridges the relationship between Gal-3 and FOXD1, and FOXD1 and Gal-3 form a positive feedback loop in lung cancer. However, we cannot rule out the possibility that Gal-3 may regulate FOXD1 through a different molecule on the cell surface.

Given the relationship between FOXD1 and Gal-3, it is most likely that targeting both FOXD1 and Gal-3 may have greater effects on tumor progression that targeting either protein alone [11,38]. In support of this possibility, co-knockdown not only resulted in a reduction of FOXD1 and Gal-3 expression but also further suppressed tumor growth and motility (Figure 5A–C). Conversely, co-upregulation resulted in the increase in the expression of these proteins (Figure 5D–F). Our findings demonstrate that this positive feedback loop between FOXD1 and Gal-3 might regulate lung cancer aggressiveness in an additive manner.

Abnormal expression of FOXD1 and Gal-3 is known to contribute to cancer pathogenesis [15,16,21], which makes them therapeutic targets for various cancers including lung cancer. However, the precise clinical pathological correlation between FOXD1 and Gal-3 remains poorly defined. In this study, we evaluated the protein levels of FOXD1 and Gal-3 in clinical human lung tumor microarrays. We and others have shown that FOXD1 or Gal-3 are increased in advanced tumor tissues (Figure 6A–B) [15,16,21]. Based on the score of the tissues, we provided new evidence that the FOXD1 expression level positively correlated with Gal-3 in human lung tissues (Appendix A). These data led us to hypothesize that FOXD1 and Gal-3 form a regulatory circuit during the tumorigenesis of lung cancer. A schematic diagram is presented to illustrate the interrelationships between FOXD1 and Gal-3 (Figure 7),

## 4. Materials and Methods

### 4.1. Cell Culture

Lung cell lines were a kind gift from Michael Hsiao (Academia Sinica, Taiwan) and lung cancer cells were maintained in RPMI1640 medium (Gibco, USA) (CL1-0, CL1-5, H1299, H520, H661) or Dulbecco’s modified Eagle medium (DMEM) (Gibco, USA) (A549, PC13, PC14) and Minimum Essential Eagle Medium with Earle’s BSS (MEM Eagle EBSS) (Gibco, USA) (WI-38) containing 10% fetal bovine serum (FBS; Gibco, USA), supplemented with 100 units/mL penicillin and 100 μg/ml streptomycin. Cells were maintained in a 37 °C incubator with a humidified atmosphere containing 5% CO_2_.

### 4.2. Transfection

Cells were used from passages three to six. GenMute transfection reagent (SignaGen Laboratories, MD, USA) were used to transfect various siRNA and plasmids.

### 4.3. mRNA-Based qPCR Assay

Cell RNA was extracted using RNA extraction kit according to the manufacturer’s protocol (Invitrogen, USA). cDNA was reversed by ToolsQuant II Fast RT kit (Tools Biotech, Taiwan). The mRNA exchange fold was quantitated by the real-time RT-PCR using the Fast SYBR Green Master Mix (QIAGEN, Germany). The PCR protocol was followed by 7500 Fast Real-Time PCR System (Thermo Scientific, USA). The relative expression of the genes was normalized by GAPDH as an internal control.

### 4.4. Immunoblotting and Immunoprecipitation

Harvested cells were lysed by RIPA lysis buffer (Santa Cruz Biotech, USA) containing protease and phosphatase inhibitors, and the proteins were loaded in equal amounts into optimal sodium dodecyl sulfate-polyacrylamide gel electrophoresis (SDS-PAGE). Followed by gel electrophoresis, the separated proteins were transferred to the 0.45-μm PVDF membrane (GE Healthcare Amersham, UK). The membranes were incubated with 5% skim milk (Difco, BD, USA) to block the non-specific signal, then incubated with specific primary antibodies at 4 °C overnight. On the next day, the membrane was incubated with horseradish peroxidase (HRP)-conjugated secondary antibody at room temperature for 1 h. Peroxidase activity was detected using chemiluminescence (Perkim Elmer, USA), and the intensity was quantified by software ImagePro. For immunoprecipitation, harvested cells were immunoprecipitated with specific primary antibody or control IgG, together with protein G Mag Sepharose overnight. Collection of the immunoprecipitated pellet was done with MagRack6, and the pellet was washed with RIPA buffer, non-immunoprecipitated lysate (Input) and immunoprecipitated lysate were immunoblotted with the indicated antibody.

### 4.5. Immunohistochemistry Analysis

Clinical lung carcinoma tissue microarray slides (BC041115d, US Biomax, Germany). The slide was de-paraffinized with xylene, rehydrated with alcohol and antigen retrieval, and then blocked with 5% normal goat serum for one hour at room temperature and then incubated with the FOXD1 antibody (Aviva Systems Biology, OAAB10686, USA) or Gal-3 antibody (R&D, MAB11541, USA) at 4 °C overnight. The next day, after washing with phosphate-buffered saline (PBS) the slides were incubated with goat anti-rabbit Fab at room temperature for 30 min, and labelled with ABC reagent. Then, the slides were further stained with DAB reagent. Stained slides were scanned by a ScanscopeXT system (Aperio Technologies, Vista, CA, USA) and positively stained cells in 10 fields of each section were analyzed by ImageScope 9.1 software (Aperio Technologies, USA). To evaluate the positive FOXD1 or Gal-3 intensity in each tissue sample, the stained level was further scored as: 0 (no staining), 1 (light brown), 2 (medium brown), or 3 (dark brown) by the percentage of positive cells, and the expression results were validated by qualified pathologist.

### 4.6. Colonogenic Formation Assay

Transfected lung cancer cells (2 × 10^3^ cells/well) were growth in six-well plates. After 14 days, cells were fixed by methanol-acetic acid (7:1) for 30 mins, then visualized by 0.1% crystal violet (Sigma-Aldrich, USA), and counted using a stereomicroscope.

### 4.7. Extraction of Cytoplasma/Nucleus and Membrane Raft Proteins

Cells were transfected with or without specific cDNA or si-RNA for 48–72 h, and harvested in a RIPA lysis buffer (Santa Cruz Biotech, USA) supplemented with phosphatase and protease inhibitors. Cytoplasma/nucleus and membrane raft were obtained according to the cytoplasma/nucleus and plasma membrane protein extraction kit manufacturer’s instruction. (BioVision, Mountain View, CA, USA). GAPDH was used as a cytoplasma marker. Proliferating cell nuclear antigen (PCNA) was used as a nucleus marker. Caveolin-1 was used as a membrane raft fraction marker. Clathrin heavy chain (HC) was used as a non-raft fraction marker.

### 4.8. Chromatin Immunoprecipitation Assay

The CHIP-qPCR assay was followed by the EZ–Magna ChIP A/G kit instruction (EZ–Magna ChIP A/G kit; Millipore, Billerica, MA). Harvested cells were cross-linked by using 1% formaldehyde (Sigma-Aldrich, USA) to fix protein and DNA complexes. Cells were fragmented by sonication using Ultrasonic Disruptor UD-201 (TOMY, Japan) resulting in DNA fragments from 200 to 1000 bp and DNA samples were confirmed by electrophoresis. DNA fragments containing Protein A/G magnetic beads were immunoprecipitation by anti-FOXD1 (Pierce Thermo scientific, PA5-27142) or using IgG isotype as a control antibodies in 4 °C overnight. Using the magnetic separator to isolate the DNA, the isolated DNA was analyzed by qPCR with specific Human *LGALS3* primer (GPH1003937(-)10A; +333 bp from the transcription start site) (QIAGEN, Germany).

### 4.9. Promoter Assay

The *Gal-3* promoter sequence (2000 bp) and its mutation sequence were synthesized by Biotools (Biotools, Taiwan) and cloned into pGL3 luciferase reporter vector (Promega, Madison, WI, USA). For the luciferase reporter assay, optimal numbers of lung cancer cells were transfected with specific target gene plasmid and pRL-TK plasmid (Promega, USA) as an internal control for 48 h. The luciferase activities were measured with Dual-Luciferase Reporter Assay System (Promega, USA) using a microplate reader (BioTek EL×800, USA).

### 4.10. Proliferation Assay

Transfected lung cancer cells (1 × 10^4^ cells/well) were re-plated in a 96-well plates. We discarded the supernatant, and CCK-8 solution (Dojindo, Japan) was added to each well followed by incubation for 1 h. Eventually, the absorbance of formazan crystals colored solution was measured at 450 nm by a spectrophotometer (BioTek EL×800, USA).

### 4.11. Transwell Migration and Invasion Assay

Optimal numbers of CL1-0 and CL1-5 suspended in 50–100 μL of serum-free medium were seeded into the upper chamber with a 8 μm porosity polycarbonate membrane inserts (Corning, USA), which were precoated with fibronectin (20 ng/mL; Millipore, Germany) for migration assays or with matrix gel (1 mg/mL; BD Bioscience, USA) for invasion assays. The lower chamber was contained with complete growth medium supplemented with 10% FBS as the chemoattractant. During the 16-h incubation, cells migrated through the membrane from the upper to the lower side, followed by staining with Liu’s stain A and B reagent (TonYar Biotech, R.O.C.) and counted under a microscope at 200× magnification. Experiments were performed in triplicate, and three separate experiments were conducted for each group.

### 4.12. Immunofluorescence Staining

A total of 1 × 10^4^ cells were seeded and grown in 5-mm glass slides for 24 h and subsequently fixed with methanol for 1 h at room temperature. The fixed cells were washed with PBS buffer and stained with a specific primary antibody for 24 h. To distinguish the different primary antibodies, secondary antibodies were label with fluorescence (green and red) for 2 h at room temperature. The nuclei were staining with DAPI. The images were visualized by confocal microscopy (Zeiss LSM 780 + ELYRA, Carl Zeiss, Jena, Germany) and analyzed using ZEN software.

### 4.13. In-Silicon mRNA Microarrays 

The cells were transfected with a specific cDNA vector or siRNA. RNA was extracted as follows using an RNA extraction kit. An mRNA microarray was performed by Genomics Research Center (GRC) core facility (Academia sinica, Taiwan). Transcriptional profiling data were collected from the GEO database (GSE7670) and from microarray analysis of cells with *FOXD1/Gal-3* overexpression or *FOXD1/Gal-3* knockdown. Then, the expression level of genes was further subjected to GeneSpring GX11 software (Agilent Technologies, Palo Alto, CA, USA) to normalize and generate log2 values. The genes that exhibited a more than 5-fold change were analyzed by Ingenuity Pathway Analysis software (IPA; QIAGEN, Valencia, CA, USA) to identify the upstream regulators and kinases that were responsive to FOXD1/Gal-3 overexpression or knockdown.

### 4.14. Statistical Analysis

All statistical comparisons were made using the two-tailed Student’s *t*-test. Tissue slides were scanned and analyzed using ImageScope 9.1 software (Aperio Technologies, USA). Statistical significance was evaluated by the *t* test at *, *p* < 0.05; *p* < 0.01; ***, *p* < 0.001 to represent differences between groups.

## 5. Conclusions

In summary, this study reports a novel regulatory loop between FOXD1 and Gal-3. FOXD1 and Gal-3 positively correlated with aggressiveness in human lung cancer. FOXD1 acted as a transcription factor and directly bound the promoter region of Gal-3, and FOXD1 translocation was regulated by ERK signaling. In turn, Gal-3 regulated FOXD1 activity through the ITGβ1/ERK/ETS-1 pathway. More importantly, we showed that the positive regulatory loop of activation of FOXD1 and Gal-3 increased the growth and motility of lung cancer. Moreover, FOXD1 combined with Gal-3 served as an independent prognostic factor for lung cancer patients. Altogether, our results provide new insights for developing a new therapeutic strategy by targeting the FOXD1/Gal-3 axis.

## Figures and Tables

**Figure 1 cancers-11-01897-f001:**
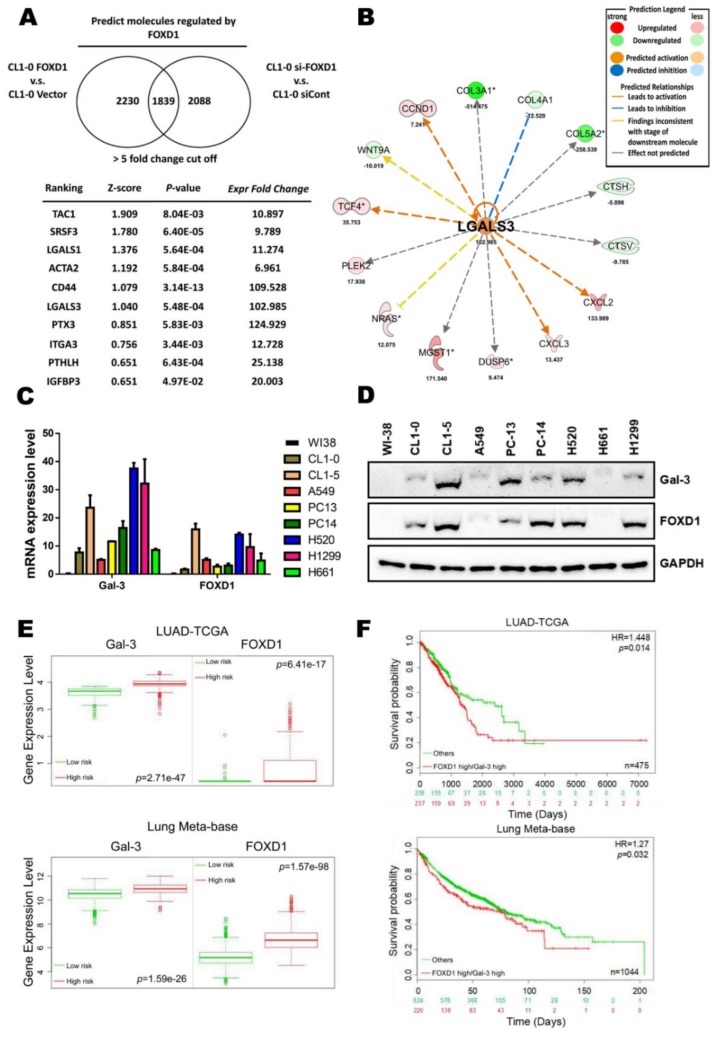
High FOXD1 and Gal-3 expression are associated with the poor prognosis in lung cancer. (**A**) The ranking of candidate molecules and the corresponding z-score and *p*-value using the Ingenuity Pathway Analyses (IPA) database from a microarray experiment performed with CL1-0 *FOXD1* compared to CL1-0 vector and CL1-5 scramble compared to CL1-5 siRNA targeting *FOXD1* (si-*FOXD1*) using a 5-fold change cut-off. (**B**) Statistical *p*-values and fold change of the expression of *Gal-3* (*LGALS3*) and its relative molecules are shown. Prediction legend: upregulation (red round icon), downregulation (green round icon), predicted activation (brown round icon), predicted inhibition (blue round icon), leads to activation (brown line), leads to inhibition (blue line), findings inconsistent with stage of downstream molecule (yellow line), effect not predicted (gray line). (**C**) RT-qPCR analysis of *Gal-3* and *FOXD1* mRNA expression in various lung cancer cell lines. *GAPDH* was used as an internal control for mRNA loading. (**D**) Western blot analysis of Gal-3 and FOXD1 protein expression in various lung cancer cell lines. GAPDH was used as an internal control for protein loading. (**E**) Lung cancer patients with high gene level transcription of *Gal-3* or *FOXD1* expression level show a correlation with high risk and (**F**) poor disease-free survival. The data were retrieved and analyzed from the TCGA-LUAD samples (*n* = 475) and lung meta-base (*n* = 1044) of the SurvExpress database (http://bioinformatica.mty.itesm.mx:8080/Biomatec/SurvivaX.jsp).

**Figure 2 cancers-11-01897-f002:**
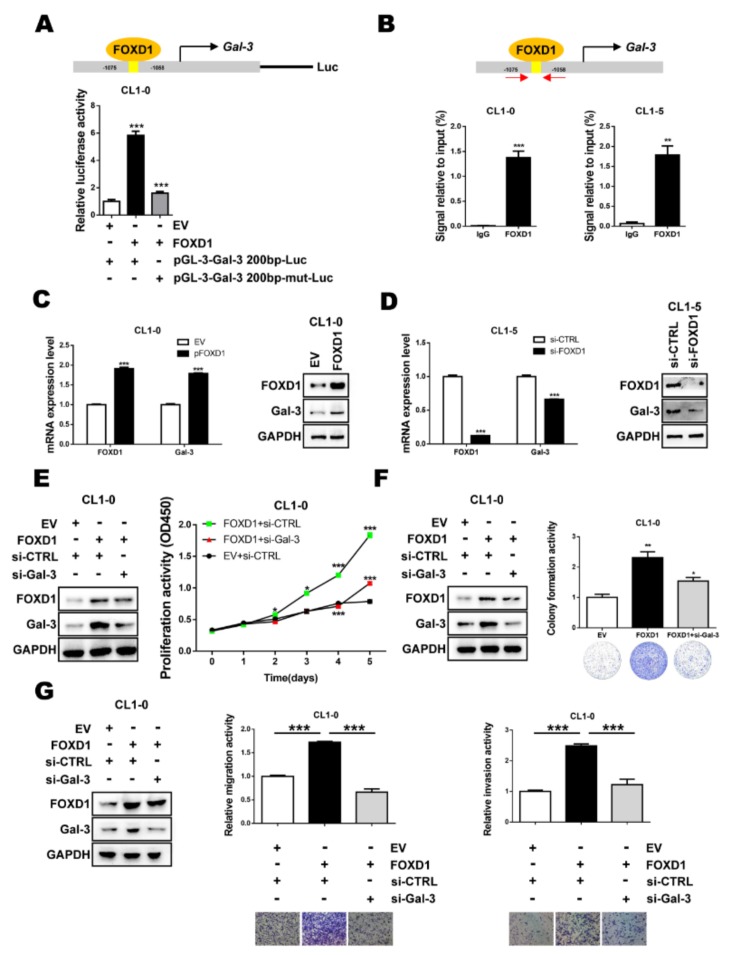
FOXD1 is a transcription factor of *Gal-3.* (**A**) CL1-0 cells were co-transfected with plasmids of the *Gal-3* promoter reporter (pGL3-*Gal-3* 2000 bp) or *Gal-3* promoter mutation reporter (pGL3-*Gal-3* 2000 bp-mut) and *FOXD1* (*pFOXD1*). The cell lysates were collected, and the promoter activity of *Gal-3* was analyzed by luciferase assay. (**B**) ChIP-qPCR assay using IgG as a control or FOXD1 antibody in CL1-0 and CL1-5 showed the binding of FOXD1 on the *Gal-3* promoter. (**C**) CL1-0 cells transfected with the empty vector or plasmid of *FOXD1* (*pFOXD1*) for 48 h and (**D**) CL1-5 cells transfected with scramble or siRNA targeting *FOXD1* (si-*FOXD1*) for 72 h were collected. The mRNA and protein levels of *FOXD1* and *Gal-3* were analyzed by qPCR (left) and immunoblotting (right), respectively. (**E**) CCK-8 assay in different groups. FOXD1-stimulated CL1-0 proliferation ability was reduced by knocking down *Gal-3*. (**F**) The colony-formation ability of FOXD1-stimulated CL1-0 cells was suppressed by the depletion of Gal-3. (**G**) The numbers of migrating or invading cells were increased in the FOXD1 group compared with the FOXD1-stimulated CL1-0-si*Gal-3* group and control group.

**Figure 3 cancers-11-01897-f003:**
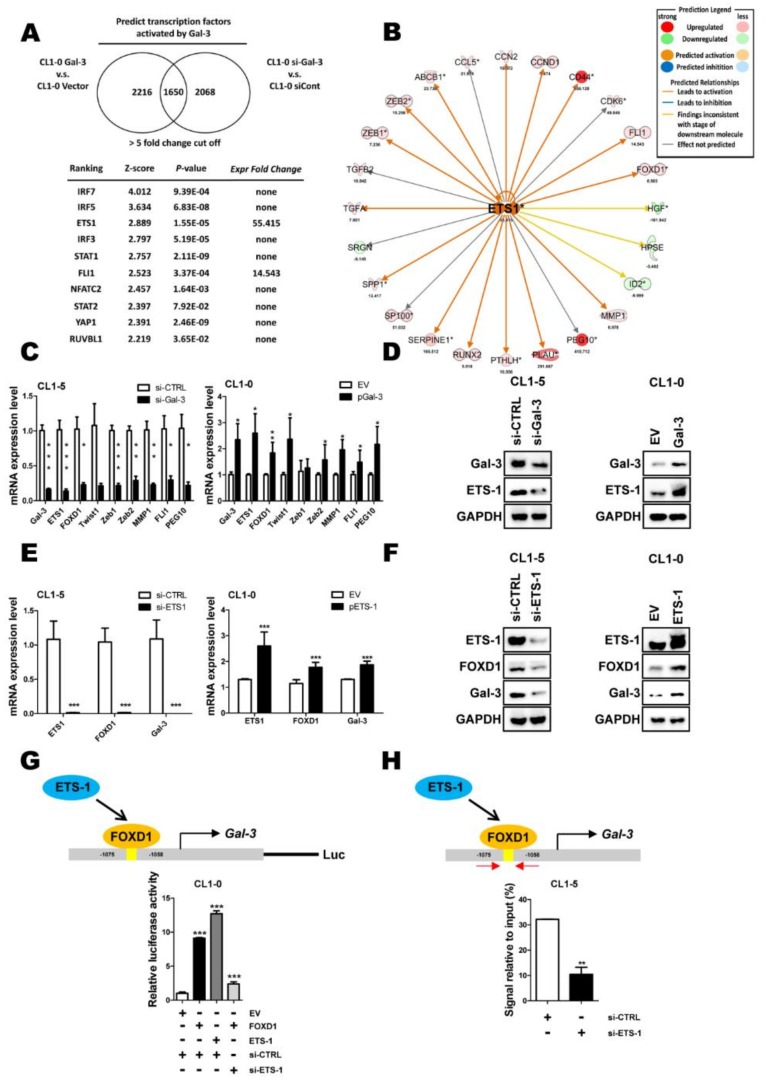
Proto-oncogene 1 (ETS-1) is involved in Gal-3-mediated FOXD1 expression. (**A**) The ranking of candidate transcription factors and its corresponding z-score and *p*-value using the IPA database from a microarray experiment performed with CL1-0 *Gal-3* compared with the CL1-0 vector and CL1-5 scramble compared with CL1-5 si-*Gal-3* using a 5-fold change cut-off. (**B**) Statistical *p*-values and fold change of the expressions of *ETS-1* (*ets-1*) and its relative molecules are shown. Prediction legend: upregulation (red round icon), downregulation (green round icon), predicted activation (brown round icon), predicted inhibition (blue round icon), leads to activation (brown line), leads to inhibition (blue line), findings inconsistent with stage of downstream molecule (yellow line), effect not predicted (gray line). (**C**) CL1-0 and CL1-5 cells were transfected with specific cDNA for 48 h or siRNA for 72 h; RT-qPCR analysis of *ets-1,* which targets downstream mRNA expression in CL1-0 cells. (**D**) Western blot analysis of Gal-3 and ETS-1 from CL1-0 with expression of the vector control (VC) or exogenous *Gal-3* gene and CL1-5 cells with scramble and siRNA targeting *Gal-3* (*si-Gal-3*). (**E**, **F**) CL1-0 and CL1-5 cells were transfected with the empty vector or plasmid of *Gal-3* (*p**Ets-**1*) for 48 h or scramble and siRNA targeting *ETS-1* (*si-**ets-1*) for 72 h, respectively, as indicated. The mRNA and protein levels of ETS-1, FOXD1, and Gal-3 were analyzed by qPCR and immunoblotting, respectively. Cells were transfected with scramble and siRNA targeting *Ets-**1* for 72 h, or empty vector or plasmid of *Ets-**1* for 48 h, respectively, as indicated, and immunoprecipitation was analyzed by immunoblotting. Input, the whole cell lysates. (**G**) CL1-0 cells were co-transfected with plasmids of *Gal-3* promoter reporter (pGL3-*Gal-3* 2000 bp), empty vector or plasmid of *FOXD1* (*p**FOXD1*) or scramble and siRNA targeting *ETS-1* (*si-**ets-1*). Cell lysates were collected, and the promoter activity of *Gal-3* was analyzed by luciferase assay. (**H**) CL1-5 cells were transfected with scramble and siRNA targeting *ETS-1* (*si-**ets-1*). ChIP-qPCR assay using IgG as a control or FOXD1 antibody in CL1-5 showed the binding of FOXD1 on the *Gal-3* promoter.

**Figure 4 cancers-11-01897-f004:**
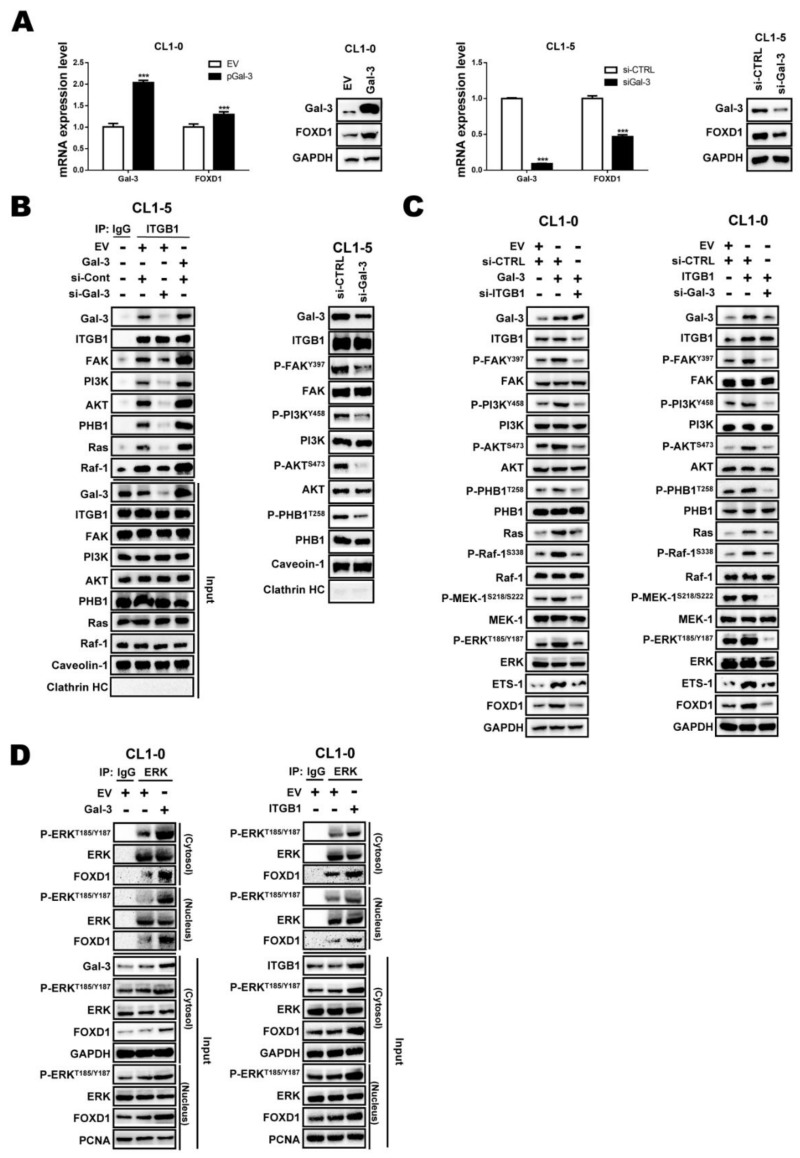
Gal-3 regulates FOXD1 expression through integrin-β1 (ITGβ1) signaling. (**A**,**B**) CL1-0 and CL1-5 cells were transfected with the empty vector or plasmid of *Gal-3* (*pGal-3*) for 48 h or scramble and siRNA targeting *Gal-3* (*si-Gal-3*) for 72 h, respectively, as indicated. The mRNA and protein levels of FOXD1 and Gal-3 were analyzed by qPCR and immunoblotting. CL1-5 cells were transfected with scramble and siRNA targeting *Gal-3* for 72 h or empty vector or plasmid of *Gal-3* for 48 h, respectively, as indicated, and immunoprecipitation was analyzed by immunoblotting. Input, the whole cell lysates. (**C**) CL1-0 cells were transfected with scramble, *Gal-3*, *ITGβ1* or si-*Gal-3,* and si-*ITG*β*1* for 72 h, as indicated. Cells lysates were collected and analyzed by immunoblotting. (**D**) CL1-0 cells were transfected with *Gal-3* or *ITGβ1* expression vector for 48 h, and immunoprecipitation was analyzed by immunoblotting.

**Figure 5 cancers-11-01897-f005:**
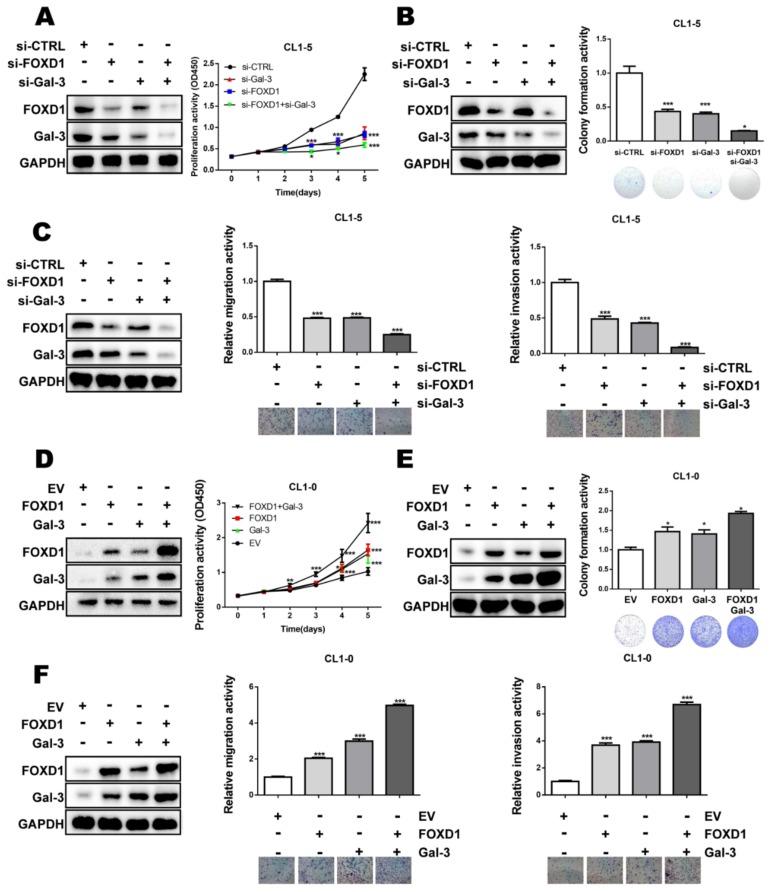
FOXD1 and Gal-3 form a positive regulatory loop to promote lung cancer cell growth and motility. CL1-5 cells were transfected with scramble, *si-FOXD1*, or *si-Gal-3* or *si-FOXD1* with *si-Gal-3* for 72 h. Cells lysates were collected and analyzed by immunoblotting. (**A**) The proliferation ability was analyzed by CCK-8. (**B**) Representative images of the colony-forming assay. (**C**) Representative images of the migration and invasion assay. CL1-0 cells were transfected with the empty vector, *FOXD1*, *Gal-3*, or both *FOXD1* and *Gal-3* expression vector for 48 h. (**D**) Proliferation ability was analyzed by CCK-8 assay. (**E**) Representative images of colony-forming assay. (**F**) Representative images of migration and invasion assay.

**Figure 6 cancers-11-01897-f006:**
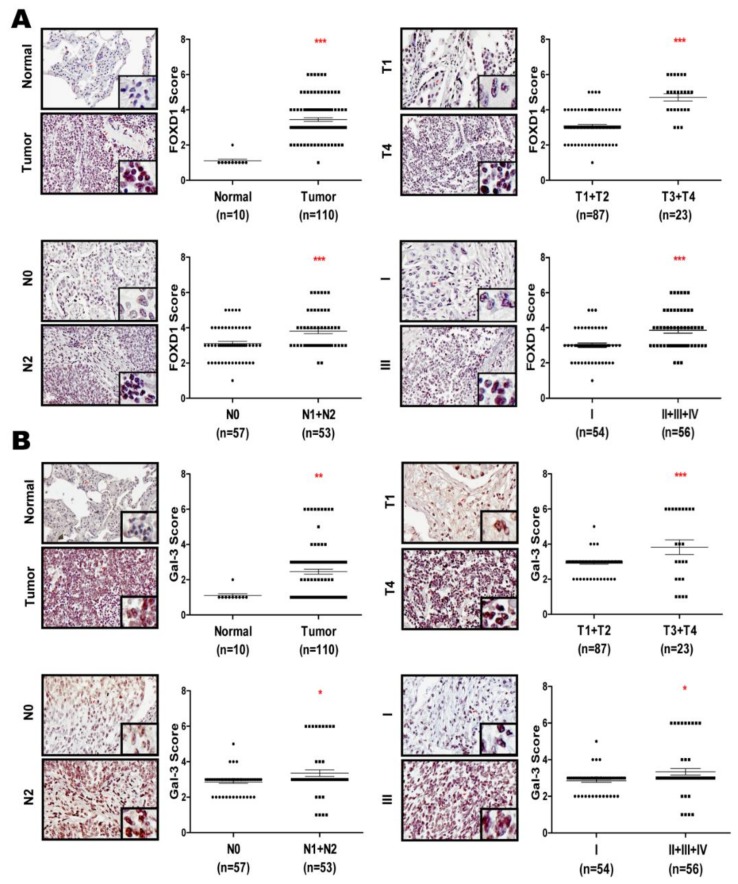
FOXD1 and Gal-3 are positively correlated in human lung cancer tissues. (**A**) Immunohistochemistry images show that FOXD1 was increased in human lung cancer tissue microarray. (**B**) Immunohistochemistry images show that Gal-3 was highly expressed in human lung cancer tissue microarray.

**Figure 7 cancers-11-01897-f007:**
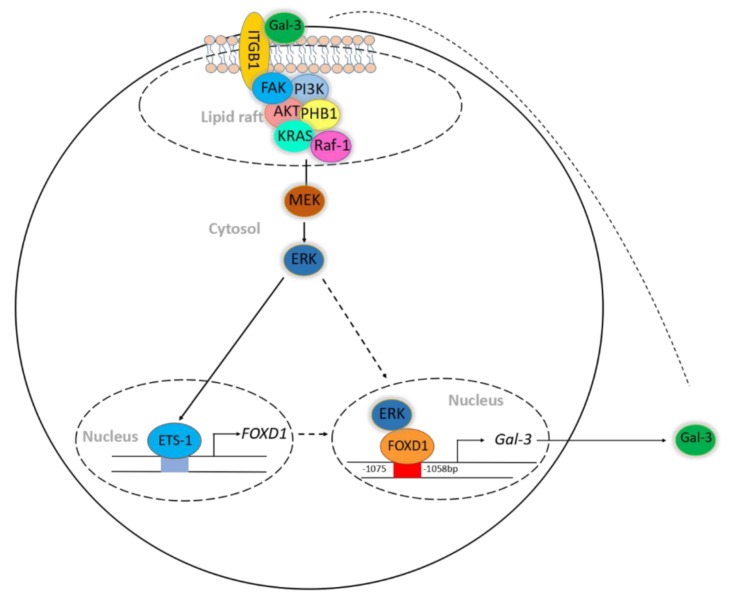
Schematic diagram illustrates cooperation between FOXD1 and Gal-3 in promoting lung cancer proliferation and metastasis.

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
