# Peer review of "FOXD1 and Gal-3 Form a Positive Regulatory Loop to Regulate Lung Cancer Aggressiveness"

_cancers, 2019, doi:10.3390/cancers11121897_

Round 1

Reviewer 1 Report

The manuscript by Chien-Hsiu Li and colleagues aims to explain the mechanism of transcription factor FOXD1 has in malignancy of lung cancers. They use genomic methods to pinpoint its partners and concentrate on the connection between FOXD1 and Gal-3 where they show that these partners interact and are involved in the positive feedback loop which is involved in the transforming the tumors more aggressive. Moreover, they show that that these players act synergistically and that their downregulation leads to less aggressive tumors proposing them as potentially good targets for therapy.

I consider that the experiments and the analysis are well performed and that the conclusions made are valid. However, some parts of the manuscript require minor adjustments (see below) and if these are corrected I would recommend the manuscript to be accepted.

Minor comments:

Fig 1A and 3A. Although the experiment is justified, there is no complete list of overlapping genes (or all genes from Venn diagrams). Another question raised is why the authors used such a high threshold (5-fold)? Moreover, it is not completely clear which gene lists are being overlapped. The ones going up in both cases? down? different directions or something else? These information should be clearly stated in the Figure legend or the main text. Results subtitle: „2.1. Gal-3 is potentially involved in FOXD1-mediated lung cancer aggressiveness.“ The title should state what was discovered/measured or describe the data obtained and not speculate with „potentially“. The authors should replace it with a different statement, for example „High FOXD1 and Gal-3 expression are associated with the poor prognosis in Lung cancer“ Fig 1B and 3B: Prediction legend is hardly readable; Moreover Figure legend for B states Log2 values are displayed in the panel, but main text states 100fold, and the number in the panel is 102.985 which confirms main text but is contrary to the legend, log2 value of 7 is 128 fold, so if it is 100 fold, Log 2 value should be less than 7, or just remain with fold expression. Fig 1C: y-axis title should state: „Relative gene expression“ or mRNA expression. word “fold” is redundant here. Also the panel would be easier to read in color, and color is already used throughout the manuscript. Fig 1E: green and red colors are supposedly low and high expression of the gene of interest (small legend within the panel), but these data are actually from low and high risk groups, as it is stated below the boxplots (x-axis). The coloring can stay but explanation should be removed as it is obvious from the y-axis which has lower or higher expression. Fig 2A: word “fold” is unnecessary as the word “relative” already contains that info (see point about Fig 1C (this is repeatedly used in figures and should be corrected throughout) Fig 2G, last panel invasion should be written, and not invation, and see remark for 2A Line 131, one “prediction” is redundant Fig S2B and statement in main text lines 133-134 are not really corroborated by the fluorescent microscopy. There is not enough evidence here to state co-localization. Both ERK and FOXD1 are indeed found in both the nucleus and the cytosol, but to claim co-localization the authors need better pictures and alternative experiments. I suggest rephrasing the statement to something like: “Both X and Y localize to the cytosol and the nucleus” Fig S2C and E. the expression of p-ERK in CL1-5 cells is lacking in both cytosol and the nucleus with the inhibitor (E) but there is some expression in the whole cell lysate. Could the authors comment on this discrepancy? Fig 3 – see some remarks above that are commented on Fig 1. which are repeated here

Author Response

Please see the attachment (Figures)

Detailed point-by-point reply (blue) to reviewer’s comments (bold).

Reviewer #1

Comments and Suggestions for Authors

The manuscript by Chien-Hsiu Li and colleagues aims to explain the mechanism of transcription factor FOXD1 has in malignancy of lung cancers. They use genomic methods to pinpoint its partners and concentrate on the connection between FOXD1 and Gal-3 where they show that these partners interact and are involved in the positive feedback loop which is involved in the transforming the tumors more aggressive. Moreover, they show that that these players act synergistically and that their downregulation leads to less aggressive tumors proposing them as potentially good targets for therapy.

I consider that the experiments and the analysis are well performed and that the conclusions made are valid. However, some parts of the manuscript require minor adjustments (see below) and if these are corrected I would recommend the manuscript to be accepted.

 Minor comments:

Fig 1A and 3A. Although the experiment is justified, there is no complete list of overlapping genes (or all genes from Venn diagrams). Another question raised is why the authors used such a high threshold (5-fold)? Moreover, it is not completely clear which gene lists are being overlapped. The ones going up in both cases? down? different directions or something else? These information should be clearly stated in the Figure legend or the main text. Answer: We have appreciate the critical comments and suggestions from the referee. In the revised manuscript, we have included data as a reference to show the complete list of overlapping genes from GeneSpring. (please see the supplementary file named Table S1-FOXD1 microarray complete list of overlapping genes or Table S2-Gal-3 microarray complete list of overlapping genes in the revised manuscript). As to the question regarding why we used a 5-fold threshold to evaluate and select Gal-3 or ETS-1 as our downstream target, we thank the referee for pointing this out. We agree with the referee that 5-fold is a high threshold. However, in this study, we found that CL1-0 or CL1-5 cells were easier to transfect with plasmid or si-RNA. Using the 5-fold threshold can narrow down gene numbers within 2000 and then analyzed by Ingenuity Pathway Analysis (IPA). We think it is more suitable for further analysis. We thus used 5-fold to perform our analysis in this study.

Results subtitle: „2.1. Gal-3 is potentially involved in FOXD1-mediated lung cancer aggressiveness.“ The title should state what was discovered/measured or describe the data obtained and not speculate with „potentially“. The authors should replace it with a different statement, for example „High FOXD1 and Gal-3 expression are associated with the poor prognosis in Lung cancer“

Answer: We thank the referee for pointing this out, and thanks for your kind suggestion. We have followed the advice to replace the title as you mention, “High FOXD1 and Gal-3 expression are associated with poor prognosis in Lung cancer.” (please see the page2, line80 subtitle 2.1 in the revised manuscript)

Fig 1B and 3B: Prediction legend is hardly readable; Moreover Figure legend for B states Log2 values are displayed in the panel, but main text states 100fold, and the number in the panel is 102.985 which confirms main text but is contrary to the legend, log2 value of 7 is 128 fold, so if it is 100 fold, Log 2 value should be less than 7, or just remain with fold expression.

Answer: We thank the referee for the suggestion and apologize for any inconvenience. We have exchange another high-resolution prediction legend as can as possible. (please see page 4 and 9, Fig-1B and Fig-3B) In addition, we also included the following sentences to correct the word “Log2” in the figure legend.

(please see page 4, line 152-157, Fig-1B legend) of the revised manuscript: “(B) Statistical P values and fold change of the expression of Gal-3 (LGALS3) and its relative molecules were shown. Prediction legend: upregulation (Red round icon), downregulation (Green round icon), predicted activation (Brown round icon), predicted inhibition (Blue round icon), leads to activation (Brown line), leads to inhibition (Blue line), findings inconsistent with stage of downstream molecule (Yellow line), effect not predicted (Gray line).”

(please see page 9-10, line 320-325, Fig-3B legend) of the revised manuscript: “(B) Statistical P values and fold change of the expressions of ETS-1 (ets-1) and its relative molecules were shown. Prediction legend: upregulation (Red round icon), downregulation (Green round icon), predicted activation (Brown round icon), predicted inhibition (Blue round icon), leads to activation (Brown line), leads to inhibition (Blue line), findings inconsistent with stage of downstream molecule (Yellow line), effect not predicted (Gray line).”

 Fig 1C: y-axis title should state: „Relative gene expression“ or mRNA expression. word “fold” is redundant here. Also the panel would be easier to read in color, and color is already used throughout the manuscript. 

Answer: We thank the referee for pointing this out. We have followed the advice to retype all the y-axis title without the "fold". (please see the figures {Fig-1C,Fig-2A,Fig-2C,Fig-2D,Fig-2F,Fig-2G,Fig-3C,Fig-3G,Fig-4A,Fig-5B,Fig-5C,Fig-5E,Fig-5F,Fig-S3E,Fig-S3F,Fig-S3G,Fig-S3H,Fig-S5D,Fig-S5E,Fig-S6C,Fig-S6D,Fig-S6E,Fig-S6F,Fig-S6C,Fig-S6D,Fig-S6E,Fig-S6F} in the revised manuscript)

Fig 1E: green and red colors are supposedly low and high expression of the gene of interest (small legend within the panel), but these data are actually from low and high risk groups, as it is stated below the boxplots (x-axis). The coloring can stay but explanation should be removed as it is obvious from the y-axis which has lower or higher expression. 

Answer: We thank the referee for pointing this out. We apologize for the confusion. We have followed the advice to replace another figure with only the coloring, as you mention. (please see page 4, figure 1E in revised manuscript)

Fig 2A: word “fold” is unnecessary as the word “relative” already contains that info (see point about Fig 1C (this is repeatedly used in figures and should be corrected throughout)

Answer: We agree with the referee. We have followed the advice to delete the word of all “fold” in our figures according to the recommendations. (please see the figures {Fig-1C,Fig-2A,Fig-2C,Fig-2D,Fig-2F,Fig-2G,Fig-3C,Fig-3G,Fig-4A,Fig-5B,Fig-5C,Fig-5E,Fig-5F,Fig-S3E,Fig-S3F,Fig-S3G,Fig-S3H,Fig-S5D,Fig-S5E,Fig-S6C,Fig-S6D,Fig-S6E,Fig-S6F,Fig-S6C,Fig-S6D,Fig-S6E,Fig-S6F} in the revised manuscript)

Fig 2G, last panel invasion should be written, and not invation, 

Answer: We thank the referee for pointing this out and apologize for the typographical errors. “invation” has been corrected to “invasion” (please see the figures {Fig-2G,Fig-5C,Fig-5F,Fig-S3G,Fig-S3H,Fig-S5E,Fig-S6E,Fig-S6F,Fig-S7E,Fig-S7F} in the revised manuscript)

and see remark for 2A Line 131, one “prediction” is redundant 

Answer: We thank the referee for pointing this out. We have thus deleted the word “prediction” at the end of the sentences: we used Group-based prediction system (GPS) , (please see the Page 5, line 187 in the revised manuscript).

Fig S2B and statement in main text lines 133-134 are not really corroborated by the fluorescent microscopy. There is not enough evidence here to state co-localization. Both ERK and FOXD1 are indeed found in both the nucleus and the cytosol, but to claim co-localization the authors need better pictures and alternative experiments. I suggest rephrasing the statement to something like: “Both X and Y localize to the cytosol and the nucleus” Fig S2C and E.

Answer: We thank the referee for pointing this out. We have corrected this sentence in page 5,2st paragraph line 189-190. The following sentences have been inserted: “Immunofluorescence staining revealed that both ERK and FOXD1 localized to the cytosol and the nucleus of CL1-5 cells “ as you mentioned.

the expression of p-ERK in CL1-5 cells is lacking in both cytosol and the nucleus with the inhibitor (E) but there is some expression in the whole cell lysate. Could the authors comment on this discrepancy?

Answer: We thank the referee for pointing this out. We have carefully rechecked the levels of p-ERK in each image of Fig-S2C and Fig-S2E, as requested by the referee and found that these were mainly due to low exposure time of the figures. Here, we replaced Fig-S2E with the figure of a duplicate experiment (another repeats-2) that exposed longer time for publication. (please see the supplementary file, Page 6, Fig-S2E in the revised manuscript)

Fig 3 – see some remarks above that are commented on Fig 1. which are repeated here 

Answer: We agree with the referee. We have followed the advice to change the word of all “fold” in our figures according to the recommendations. (please see the figures {Fig-1C,Fig-2A,Fig-2C,Fig-2D,Fig-2F,Fig-2G,Fig-3C,Fig-3G,Fig-4A,Fig-5B,Fig-5C,Fig-5E,Fig-5F,Fig-S3E,Fig-S3F,Fig-S3G,Fig-S3H,Fig-S5D,Fig-S5E,Fig-S6C,Fig-S6D,Fig-S6E,Fig-S6F,Fig-S6C,Fig-S6D,Fig-S6E,Fig-S6F} in the revised manuscript)

Reviewer 2 Report

This paper by Li and colleagues has reported a novel regulatory loop between FOXD1 and Gal-3; they had demonstrated that FOXD1 promoted lung cancer aggressiveness by targeting Gal-3. This article provides new sights for developing a new therapeutic strategy. This is a worthwhile study, as there is a significant need for novel treatment for patients with lung cancer. This paper is well written; study is well designed. I will suggest following modifications:

In the background section: to be precise when mentioning about the management of lung cancer, based on staging. In line 36, page #1, please mention the 5- year survival in figure rather than saying low. It might be worthwhile to mention about the response rate and the md PFS and OS in the second line settings in the background section. Page #4, figure b: hard to read the content. Similar problem in page #9, figure B. I will suggest mentioning figure #7 in the discussion section and not in the conclusion.

Author Response

Please see the attachment (Figure)

Detailed point-by-point reply (blue) to reviewer’s comments (bold).

Comments and Suggestions for Authors

This paper by Li and colleagues has reported a novel regulatory loop between FOXD1 and Gal-3; they had demonstrated that FOXD1 promoted lung cancer aggressiveness by targeting Gal-3. This article provides new sights for developing a new therapeutic strategy. This is a worthwhile study, as there is a significant need for novel treatment for patients with lung cancer. This paper is well written; study is well designed. I will suggest following modifications:

Inthe background section: to be precise when mentioning about the management of lung cancer, based on staging. In line 36, page #1, please mention the 5- year survival in figure rather than saying low. It might be worthwhile to mention about the response rate and the md PFS and OS in the second line settings in the background section.

Answer: We are appreciate the critical comments and kind reminder from referee.  We have included the following sentences into the introduction section to describe as more precise as we can. (please see page 1, introduction section, 1st paragraph, line 35-39) of the revised manuscript: “However, patients with advanced lung cancer have 10-15% overall 5-year survival rates, the median overall survival (OS) about 11.1 months, the post-recurrence survival rate (PRS) was 13% [hazard ratio (HR)=0.78] [2-4]. Even the second-line drug treatment, such as Nivolumab, the progression-free survival (PFS) about 2.3-3.5 months [5].” 

Page #4, figure b: hard to read the content. Similar problem in page #9, figure B.

Answer: We thank the referee for point this out and apologize for any inconvenience. We have replaced it with another high-resolution prediction legend. (please see page 4 and 9, Fig-1B and Fig-3B)

(please see page 4, line 152-157, Fig-1B legend) of the revised manuscript: “(B) Statistical P values and fold change of the expression of Gal-3 (LGALS3) and its relative molecules were shown. Prediction legend: upregulation (Red round icon), downregulation (Green round icon), predicted activation (Brown round icon), predicted inhibition (Blue round icon), leads to activation (Brown line), leads to inhibition (Blue line), findings inconsistent with stage of downstream molecule (Yellow line), effect not predicted (Gray line).”

(please see page 9-10, line 320-325, Fig-3B legend) of the revised manuscript: “(B) Statistical P values and fold change of the expressions of ETS-1 (ets-1) and its relative molecules were shown. Prediction legend: upregulation (Red round icon), downregulation (Green round icon), predicted activation (Brown round icon), predicted inhibition (Blue round icon), leads to activation (Brown line), leads to inhibition (Blue line), findings inconsistent with stage of downstream molecule (Yellow line), effect not predicted (Gray line).”

I will suggest mentioning figure #7 in the discussion section and not in the conclusion. 

Answer: We thank the referee for pointing this out. We have followed the advice and move the figure-7 to the discussion section. (please see the page 17, 3rd paragraph, line 585-586, page 17, and page18, figure 7 in revised manuscript)

Reviewer 3 Report

The paper by Chien-Hsiu Li et al. titled “FOXD1 and Gal-3 form a regulatory loop…” is an interesting search for the possible links between the action of transcription factor FOXD1 and lectin family member. From this study it seems obvious that FOXD1 and Gal-3 form a feed-forward loop which might explain the progressive resistance to chemotherapy in non-small cell lung carcinoma (NSCLC). This paper needs a list of abbreviations, since a number listed in the text is incomprehensible to the average reader.

Author Response

Please see the attachment (Figure)

Detailed point-by-point reply (blue) to reviewer’s comments (bold).

Reviewer 3

Comments and Suggestions for Authors

The paper by Chien-Hsiu Li et al. titled “FOXD1 and Gal-3 form a regulatory loop…” is an interesting search for the possible links between the action of transcription factor FOXD1 and lectin family member. From this study it seems obvious that FOXD1 and Gal-3 form a feed-forward loop which might explain the progressive resistance to chemotherapy in non-small cell lung carcinoma (NSCLC).

This paper needs a list of abbreviations, since a number listed in the text is incomprehensible to the average reader.

Answer: We appreciate the critical comments and kind reminder from referee. We have followed the advice. In the “Abbreviations” section of the revised manuscript (please see page 21, abbreviations section, line 758-772), the following sentences have been inserted.

Abbreviations:

ETS-1     Proto-oncogene 1

FOXD1    Forkhead box D1

FOX      Forkhead box

Gal-3      Galectin-3/LGALS3

GPS       Group-based prediction system

HR        Hazard ratio

ITGB1     Integrin-β1

IPA       Ingenuity Pathway Analysis

NSCLC    Nonsmall cell lung cancers

NSL       Nuclear localization signals

OS      Overall survival

PFS     Progression-free survival

PRS     Post-recurrence survival rate

TF        Transcription factor

Reviewer 4 Report

Li et al. present an excellent study determining the role of FOXD1/Gal-3 on tumor cell growth promotion. Their study is a thorough work showing a comprehensive set of experiments determining that the axis ITGB1/ERK/FOXD1/Gal-3 is contributing to tumor malignancy. On top of that authors show that some of the components of this axis correlate with malignancy in human patients, indicating that this could be a relevant signaling pathway.

My only minor suggestion would be to change the term synergistic in the FOXD1 and Gal-3 interaction, because the experiments shown probe that this relation is merely additive.

Author Response

Detailed point-by-point reply (blue) to reviewer’s comments (bold).

Reviewer 4

Comments and Suggestions for Authors

Li et al. present an excellent study determining the role of FOXD1/Gal-3 on tumor cell growth promotion. Their study is a thorough work showing a comprehensive set of experiments determining that the axis ITGB1/ERK/FOXD1/Gal-3 is contributing to tumor malignancy. On top of that authors show that some of the components of this axis correlate with malignancy in human patients, indicating that this could be a relevant signaling pathway.

My only minor suggestion would be to change the term synergistic in the FOXD1 and Gal-3 interaction, because the experiments shown probe that this relation is merely additive.

Answer: We thank the referee for pointing this out. In the revised manuscript, We have followed the advice. We replaced the word of all “synergistic” to “positive regulatory loop” or “in an additive manner” in the revised manuscript according to the recommendations. (please see the revised manuscript{page1, line2}{page1, line24}{page12, line383}{page12, line384}{page12, line389}{page12, line391}{page12, line392}{page12, line396}{page13, line432-433}{page17, line573}{page17, line576}{page21, line726}
